# Effectiveness of Computerized Cognitive Training by VIRTRAEL on Memory and Executive Function in Older People: A Pilot Study

**DOI:** 10.3390/brainsci13040684

**Published:** 2023-04-19

**Authors:** Sandra Rute-Pérez, Carlos Rodríguez-Domínguez, María Vélez-Coto, Miguel Pérez-García, Alfonso Caracuel

**Affiliations:** 1CIMCYC—Mind, Brain and Behavior Research Center, University of Granada, 18011 Granada, Spain; 2Department of Developmental and Educational Psychology, Faculty of Education Sciences, University of Granada, 18011 Granada, Spain; 3Department of Computer Languages and Systems, Faculty of Education, Economy and Technology of Ceuta, University of Granada, 51001 Granada, Spain; 4Department of Psychology, Catholic University of Murcia, Guadalupe, 30107 Murcia, Spain; 5Department of Personality, Assessment and Psychological Treatment, Faculty of Psychology, University of Granada, 18011 Granada, Spain

**Keywords:** computerized cognitive training, traditional paper-and-pencil cognitive training, cognitive stimulation, older adults, aging

## Abstract

The prevalence of people over 60 years of age with cognitive impairment has increased in recent decades. As a consequence, numerous computerized cognitive trainings (CCT) have been developed. This pilot study aimed to determine the effectiveness of the CCT with VIRTRAEL in improving older adults’ cognition. Fifty-five participants ($\bar{x}$ = 72.7 years; SD = 6.5) underwent CCT, and twenty participants ($\bar{x}$ = 76.1 years; SD = 7.6) received face-to-face cognitive stimulation with a paper-and-pencil methodology. Both trainings were conducted in nine sessions (45–60 min each). Participants completed a pre-post training neuropsychological assessment. ANCOVAs and the standardized clinical change were performed. VIRTRAEL’s group showed a significant and greater improvement in verbal learning (*p* < 0.006) and delayed recall (*p* ≤ 0.001), working memory (*p* < 0.005), abstract (*p* < 0.002) and semantic reasoning (*p* < 0.015), and planning (*p* < 0.021). Additionally, more large clinical changes (*d* > 0.8) were found in the VIRTRAEL condition (in verbal learning and delayed free and cued recall) than in the standard group. Here we show that the CCT with VIRTRAEL is effective in improving cognitive function in older adults and is superior to the standard format. These preliminary findings indicate that CCT is a useful tool potentially applicable in the fight against cognitive symptomatology associated with aging and neurodegenerative diseases. VIRTRAEL represents a breakthrough in this field as it is inexpensive and easily accessible to any older person, regardless of whether they live far from health care resources.

## 1. Introduction

According to the United Nations Organization (UNO) [1], the population over 60 years of age has doubled in the last four decades and is expected to reach 1.5 billion people worldwide by 2050. This aging trend entails a significant increase in the prevalence of chronic diseases such as neurodegenerative diseases, which are the fourth leading cause of mortality in older adults (5%) behind cardiocirculatory diseases (30.5%), tumor diseases (28.2%), and respiratory diseases (10.9%) [2]. Some of the main neurodegenerative diseases affecting cognitive functioning and independence in the older adults are dementia, Alzheimer’s disease (AD) [3], and mild cognitive impairment (MCI) [4], which carry a significant socioeconomic cost.

Most cases of neurodegenerative diseases are multifactorial, in which environmental (e.g., diet) and genetic (e.g., mitochondrial impairment and tau and amyloid-β deposits) risk factors interact cumulatively over an individual’s lifetime [5,6,7]. Cognitive impairments in attention, memory, and executive functions, as well as behavioral and emotional symptoms common in neurological diseases, correlate with a pattern of prefrontal lobe dysfunction. Nevertheless, the manifestation or not of symptoms in neurological and neuropsychiatric diseases is the result of the balance between the processes of neurodegeneration and neuroprotection [8]. Therefore, despite facing a decline in cognitive abilities, the human brain maintains the ability to adapt to environmental changes even at advanced ages [9]. Thus, recent studies have found that there are preventive interventions that can help maintain and improve cognitive function [10] and even reverse the cognitive deterioration in people with MCI to the extent of returning to a normal functioning [11,12,13,14]. For example, a systematic review carried out by Grande et al. in 2016 [15] found that the reversion rates ranged between 29–55% in population studies and 4–15% in clinical studies.

In recent decades, non-pharmacological interventions have reported significant scientific support as strategies for the prevention and treatment of cognitive impairment [16,17]. Cognitive stimulation or training has become the quintessential strategy for cognitive impairment due to its ability to enhance experience-dependent neuroplasticity and evidence of its effectiveness in improving performance or functionality in healthy older adults and people with MCI, dementia, or Alzheimer’s disease [18,19,20,21,22,23,24,25]. For example, in a meta-analysis [26], the results showed that traditional cognitive training using pencil and paper improved different cognitive functions such as memory, language, executive function, visuospatial skills, attention, and processing speed, among others, as well as the quality of life and depression in people with impairment.

Advances in information and communication technologies have led to the emergence of computerized cognitive training (CCT) aimed to prevent and reduce cognitive impairment [27,28,29]. Many of these programs offer a number of significant advantages [21,30]: (1) they allow an individualized approach according to the needs and characteristics of each person, (2) they are accessible to a greater number of people and avoid problems due to reduced mobility and/or access to health resources, (3) they involve a lower economic cost, and (4) they allow an objective analysis of performance and immediate feedback. Evidence from a systematic review and meta-analysis indicated that CCT improves global cognition, specific cognitive domains (verbal learning, verbal memory, nonverbal learning, working memory, and attention), and psychosocial functioning in healthy older adults and people with MCI [12,24,28,31,32,33,34,35,36,37]. These findings have encouraged many companies to develop CCT products, promising to reduce and/or prevent cognitive impairment in older adults. However, some of them exaggerate or misrepresent information about the benefits of the product [38] and lack effectiveness studies. There are some limitations that are hindering the scientific-professional development of this field. Regarding research designs, many CCTs do not have an empirical basis, but they are a mere compendium of exercises [27]; they are based on the training of a single cognitive domain [35]; and there is a lack of control or there are inactive groups in most of the studies [30,34]. In addition, as for marketing, they are usually expensive for the average older people [16].

Despite these limitations, studies point to a greater effectiveness of CCT compared to “traditional paper-and-pencil cognitive training (PCT)”. For instance, a recent study by Bernini et al. [39] found that those patients who received CCT showed significant medium/large-sized improvements in the Montreal Cognitive Assessment (MoCA) performance, global cognition, executive functions, and attention/processing speed in comparison to a PCT group and a control group. These results support previous systematic reviews, such as that one of Kueider et al. [24] where within computerized cognitive studies the mean effect sizes for certain cognitive functions were higher than in classic interventions. For instance, processing speed showed a mean effect size of 4.00 compared to a 1.30 from paper-and-pencil intervention.

To alleviate some of the above limitations, the authors of the present study received public funding and developed the VIRTRAEL computerized cognitive stimulation website (“Virtual Training for the Elderly people”, n.d.). It is a free access platform that includes tests and exercises for the assessment and stimulation of frequently altered cognitive skills in older people, such as attention, learning, memory, and executive functions. All training exercises are of increasing difficulty, with stimuli and contexts familiar to older adults. The rationale for the inclusion of activities, their frequency, sequences, and parameters, are based on paradigms and models from cognitive neuroscience [40] and neuropsychological intervention [41].

The aim of this pilot study was to determine the effectiveness of the VIRTRAEL online platform in improving cognitive status in older adults, in comparison to traditional paper-and-pencil cognitive training. We consider this work necessary to detect possible study failures or problems and to reduce the probability of wasting time, effort, and money in a clinical study with a larger population. We hypothesize that given its characteristics and previous findings, VIRTRAEL will improve performance in attention, learning, memory, and executive function beyond the improvements obtained through a standard stimulation program in older adults.

## 2. Materials and Methods

### 2.1. Design

A quasi-experimental repeated measures study was carried out between and within subjects. Seventy-five older adults were recruited, twenty of whom were randomly assigned to the traditional face-to-face condition because the capacity of that group was limited to that number by staffing and space limitations. The remaining participants were assigned to the computerized cognitive training group with VIRTRAEL.

### 2.2. Participants

Seventy-five older people participated in the study and were divided into two groups: VIRTRAEL, those who received training through the online platform, and a standard group, who received a traditional paper-and-pencil cognitive training. All participants registered for participating in a cognitive training workshop offered in a community center in the province of Granada (Andalusia, Spain).

The VIRTRAEL group was composed of 55 participants (74.5% women), between 63 and 91 years old, whose mean age was 72.7 years (SD = 6.5). The standard group consisted of 20 participants (75% women) between the ages of 63 and 88 (mean age = 76.1 years, SD = 7.6). This sample is consistent with previous studies [42] and is representative of the older population participating in cognitive stimulation programs in terms of age and sex. In addition, the real percentage of attendance at civic centers is higher for women than for men. The sociodemographic and clinical characteristics are displayed in Table 1.

The inclusion criteria were having (1) an age of more than 60 years, (2) a score ≥ 21 on the Mini-Mental State Examination, (3) no clinical signs of dementia (Reisberg Global Deterioration Scale ≤ 3), and (4) a basic level of reading and writing skills verified by the examiner. The exclusion criterion was having a medical diagnosis of dementia or any systemic condition associated with cognitive deficits.

The dropout rate was 0, as all participants were involved until the end of the study.

### 2.3. Instruments

Participants were assessed individually before and after the intervention using standardized paper-and-pencil tests.

The d2 Test of Attention [45] measures selective attention and concentration. In this study, only the concentration index was used: d2-Concentration = total number of correct responses—commission errors.

The Hopkins Verbal Learning Test-Revised (HVLT-R) [46,47], forms A and B (used in the pre and post assessment, respectively), measures learning and verbal memory. The indexes included in this study were as follows: total learning = sum of trials 1–3; delayed recall = No. correct words in the delayed free recall trial; and recognition = No. target words correctly recognized. Additionally, a delayed cued recall score was included since following the delayed free recall, participants were given recall cues related to the word categories from the list (i.e., Form A: birds, drinks, tools; Form B: fruits, gems, buildings) [48].

Letter-Number (L&N), Similarities, and Matrix sequencing subtests from the Wechsler Adult Intelligence Scale (WAIS-III used in the pre-training assessment, and WAIS-IV in the post-training) [49,50] measure working memory and semantic and abstract reasoning, respectively. The overall raw scores were used.

Lastly, the Keys Search subtest of the Behavioral Assessment of the Dysexecutive Syndrome (BADS) tests battery was used [51]. This test was designed as an ecological instrument to measure the ability of planning, as part of the executive function. The final raw score, ranging from 0 to 16 points, was used.

All instruments were validated for the cognitive assessment of older adults. Information is included in the manual of each instrument [45,46,47,48,49,50,51].

To conduct the computerized cognitive stimulation VIRTRAEL (“Virtual Training for the Elderly people”, n.d.) (http://www.everyware.es/webs/virtrael/#home) (accessed on 16 February 2023) was used. It is an online platform, freely accessible upon registration request, designed for cognitive assessment and stimulation of older people. An initial and shortened version called PESCO [38] has shown validity for the improvement of attention, working memory, and planning in older adults without dementia. That initial version was extended with more activities and its online access was improved, so that it is now available as a web platform for all browsers [52,53,54,55].

VIRTRAEL includes a total of 11 types of stimulation exercises distributed in 9 sessions of 45–60 min each, depending on the speed and quality of execution of the user. In a session, there can be 3 to 5 exercises and the level of difficulty of each exercise automatically adapts to the person, using an algorithm based on previous performance (successes, failures, completion times, etc.). The content of each session was programmed beforehand, and the included exercises were ordered to train cognitive functions (attention; verbal, visual, and working memory; reasoning; and planning) in a sequential manner (e.g., first attention, then memory). Maintenance of motivation was addressed by varying the type of exercises in each session, using an avatar, and a virtual medal-based rewarding system. In addition, the design of the exercises had an ecological approach with the objective of the performance of recalling daily life activities and facilitating the possible transfer to real life (Figure 1a,b and Figure 2a,b show examples of some exercises). The type of exercises, their combinations during the sessions, and the specific parameters for each one were established according to the descriptions of activities contained in a seminal manual on neuropsychological rehabilitation [41] and specific evidence in computer cognitive training [24]. Exercises and sessions are described in Appendix A, and in the following link it is possible to access a demo of VIRTRAEL: https://virtrael-demo.web.app/exercises (accessed on 16 February 2023)

The cognitive stimulation the standard group received consisted of the realization of a workshop that the center’s staff of psychologists developed following traditional cognitive stimulation guidelines [56,57,58]. Thus, different exercises involving attention, working memory, verbal memory, verbal fluency, reasoning, arithmetic, and planning were performed. Sessions were programmed with paper-and-pencil exercises that were ordered and had the same duration as for the sessions in VIRTRAEL.

### 2.4. Procedure

All the participants were informed about the project and signed the informed consent. Both interventions (VIRTRAEL and the standard program) were conducted in groups of approximately 10 people in sessions of 45–60 min twice a week, summing up a total of 9 sessions. Although the sessions on both conditions were in a group, each user performed their exercises individually on independent computers in the case of VIRTRAEL, and the professional in charge approached each of them if there were any questions or problems.

In the group that received the stimulation with VIRTRAEL, an initial task to train the use of the mouse was included at the beginning of the program. Participants would only begin program tasks once they had reached an appropriate level of use.

### 2.5. Statistical Analysis

Statistical analyses were performed using the IBM Statistical Package for the Social Sciences (SPSS) version 26 for Windows.

First, the means and standard deviations of the sociodemographic characteristics of the two groups were calculated. To compare the difference between both groups at the pre-intervention time point, independent-sample t-tests were performed. The groups showed significant differences in four of the dependent variables: attention (*t* = 2.960; *p* = 0.004), working memory (*t* = 2.131; *p* = 0.036), abstract (*t* = 2.369; *p* = 0.003), and semantic reasoning (*t* = 3.010; *p* ≤ 0.005). Therefore, to consider the effect of these differences between groups, all the multiple Analyses of the Covariance (ANCOVAs) were performed using the group (VIRTRAEL versus standard stimulation) as the independent variable, the score of each cognitive variable at the post-intervention time point as the dependent variable, and the score of the same cognitive variable at the pre-intervention time point as the covariate.

Secondly, the individual (within the subject) clinical change was determined. This procedure was carried out in two steps: (1) The size of the individual standard effect for each participant and each cognitive variable was calculated according to the following formula [59,60]: *δ* individual = (s2 − s1)/*σ*, where s1 = score of the individual at the beginning of the study (pre-intervention); s2 = the score of the individual at follow-up (post-intervention); and *σ* = the standard deviation of its group at the beginning of the study. (2) The effect size was obtained at the group level, calculating the mean effect size of all the individuals within each group.

## 3. Results

Table 1 shows an overview of the sociodemographic and general cognitive status of the sample. The age range was 63–91 years in the VIRTRAEL group and 63–88 years in the standard group. The groups showed no significant differences in age, educational level, cognitive reserve, and MMSE score. The percentage of women in the groups (74.5% and 75%, respectively) showed no significant difference [*X*^2^ (1,75) = 0.002; *p* = 0.968].

Levene’s test showed equality of variances in all dependent variables. ANCOVA analyses showed that the participants in the VIRTRAEL group had a significantly higher improvement than those in the standard group in learning (HVLT-R Total learning) and verbal memory (HVLT-R delayed recall and HVLT-R cued recall), working memory (L&N), abstract (Matrix) and semantic (Similarities) reasoning, and planning (Keys Search). No differences were found in attention (d2CON) and recognition memory (HVLT-R Recognition) (See Table 2).

To check the effect size of each group, a within-subject pre–post effect size for every individual in each domain of each variable was calculated, and the group mean was obtained. The average Cohen’s *d* (pre–post effect size) in the VIRTRAEL group was 0.67 (range= 0.24 to 0.95), whereas in the standard treatment group it was 0.38 (range= −0.16 to 0.84) (See Table 2). In the VIRTRAEL group, large clinical changes (*d* > 0.8) were found in HVLT-R delayed recall, HVLT-R cued recall, HVLT-R total learning, and Matrix; moderate clinical changes (*d* = 0.5) in L&N and Similarities; and small clinical changes (*d* = 0.2) in d2-Concentration, HVLT-R Recognition, and Keys Search.

In the standard group, it was found that there was only one large clinical change (*d* > 0.8) in Matrix; moderate clinical changes (*d* = 0.5) in HVLT-R Recognition and Similarities; and low clinical changes (*d* = 0.2) in d2-Concentration, HVLT-R total learning, and L&N (see Table 2) (see Figure 3).

In summary, the VIRTRAEL group showed a significant and greater improvement in verbal learning (*p* < 0.006), delayed recall (*p* ≤ 0.001), working memory (*p* < 0.005), abstract (*p* < 0.002) and semantic reasoning (*p* < 0.015), and planning (*p* < 0.021). Additionally, large clinical changes (*d* > 0.8) were found in the VIRTRAEL condition in verbal learning, delayed recall and cued with semantic keys, and abstract reasoning. In the standard group, a large clinical change was found just in abstract reasoning.

## 4. Discussion

The aim of the pilot study was to determine the effectiveness of VIRTRAEL in improving the cognitive status of older people compared to a standard cognitive stimulation program. The preliminary findings indicated that the group trained with VIRTRAEL had statistically higher scores than those of the standard in verbal learning and memory, working memory, abstract and semantic reasoning, and planning.

These findings indicate that individually constrained cognitive training through the web platform was more effective than through the traditional format of paper-and-pencil exercises provided by a professional psychologist. Back in 1989, Finkel and Yesavage [61] carried out a similar study and, although they found that both groups improved after the stimulation, there were no significant differences between them. Currently, thanks to the technological advance of computerized cognitive training programs, there is a large number of studies that have shown significant improvements [32,59,60]. However, the research approach to this field needs improvements as many of the studies that yield positive results have included inactive control groups or a waitlist [62,63,64,65], activities that do not directly involve cognitive training, such as psychoeducation [66] or pharmacological treatment, and small effect sizes [17,32] or heterogeneous results [28,31,34,37]. The ultimate goal of this study was to determine, through an appropriate research design, the benefits of participating in an online cognitive stimulation program.

After the stimulation, the groups were matched in only two cognitive domains: attention and recognition of the words of a previously presented list. Regarding attention, both groups improved in a similar way. These results coincide with the literature where the most frequent findings indicate that improvements in attention are not different between the computerized and standardized formats [24,32]. A possible explanation for this finding can be found in the great reactivity of the attention to the intervention attempts. Programs aiming to improve this cognitive function have shown greater degrees of effectiveness than those focusing on other cognitive functions, either in traditional or computerized formats, for older people [67] or those with brain damage acquired [68]. It is possible that the exercises presented in both formats are sufficiently activating the brain plasticity to improve the networks responsible for attention. Therefore, for computerized exercises to be more effective than pencil-and-paper exercises, they should be presented with parameters that are even more challenging than those currently available.

The lack of differences between the groups in the capacity for verbal memory recognition might be explained because the margin of change in this aspect was very small since both groups started from a very high score of discrimination before the training. This was expected according to the typical mnesic profile in most people [69]. Both training formats have slightly improved this high previous performance, until reaching an average score close to the maximum possible (12 correctly recognized words).

To reach the objective of the study, a determination of the clinical change that each person has experienced has also been determined, calculating the standard difference between the level reached after the stimulation and the starting level of each one. This type of data is relevant as it indicates the size of the effect produced by every stimulation format. The mean effect size for the general improvement of cognition in the older adults by computerized cognitive training is, according to a recent systematic review and network meta-analysis, 0.18 for healthy older adults [34] with considerable heterogeneity, and for individual cognitive domains, such as long-term memory and retrieval, general short-term memory, and executive function 0.16, 0.17, and 0.17, respectively. The sizes of the effect of cognitive change produced by VIRTRAEL have not only been larger than those produced by traditional stimulation (e.g., in verbal learning-memory and components of executive function) but are also larger than the average in the literature on computerized stimulation. The difference is very noticeable in learning and verbal memory, where with the computerized training large-size changes are obtained, while with the standard training, they are of small size (learning) or trivial (memory). This difference is very outstanding, considering that one of the fundamental objectives that traditional cognitive training workshops try to achieve is memory improvement, which is usually the most frequent complaint among the older adults [70]. VIRTRAEL was designed to meet the challenge of transferring the methodology of cognitive stimulation from the face-to-face format to a computerized format but trying, thanks to technology, to reinforce the strategies that have proven to be effective as much as possible. To ensure maximum effectiveness in improving memory [71], the VIRTRAEL exercises were designed so that users had to learn memory strategies, avoiding erratic memory attempts not based on strategies. Thus, participants verify from the beginning that strategies are an appropriate way of coping to overcome the challenges presented by the activities of VIRTRAEL. The design of the computerized exercises favored the person to keep the idea of the effectiveness of the application of a categorization strategy active to learn information and remember it afterwards. In contrast, in the traditional format, the application of strategies is also favored but it is impossible to control that in a group format, they are applied systematically. In a standard stimulation environment, people can or cannot apply them, and the environment cannot be controlled or programmed, which are some of the main characteristics of the computerized environment.

Regarding working memory, the difference in the size of the effect of the change found between the groups (difference of 0.3) is a relevant finding due to the centrality and prominence that this component plays within cognitive functioning, and, therefore, its repercussion on the rest of components [72]. To design VIRTRAEL, the evidence about the effectiveness demonstrated by different activities for the improvement of each cognitive function was considered. In working memory, the tasks based on the N-back paradigm were the ones with greater support [73]. In a previous study, the effectiveness of VIRTRAEL was shown to improve working memory compared to a control group that performed activities on a computer but without stimulation purposes [38]. However, many other studies have found that multidomain computer programs are also effective in improving working memory versus an active control group, with generally small or moderate effect sizes [74,75,76,77]. In this study, we show not only the benefits of the N-back paradigm for the improvement of the working memory of older people but its possibilities of being applied both in its classic format—through the Balloon exercise—and in a more ecological one in which a real context of any neighborhood is simulated—through the Bag of Items exercise.

Finally, in the planning skill, the size of the effect of the change experienced by the VIRTRAEL group after the program has been small. In most studies with a control group, no significant improvements in executive function are obtained [32]. We hypothesize that the findings of this study are due to the fact that the exercises of VIRTRAEL to train the planning were designed and subsequently programmed in a determined order within the training sessions, following an executive function model widely validated that proposes that planning is a high-level cognitive component that relies on other basic executive components such as working memory [78]. Based on this theoretical framework, VIRTRAEL designed specific planning activities, but also all the exercises necessary to train their prerequisites and distributed them synergistically throughout the sessions. The planning tasks were designed with a sufficient lack of structure so that the people had to work hard to achieve them properly. Although support elements were also included so that people could resort to the information they needed (where they could access the elements they had already completed, what they lacked, how much money they had available, etc.). In this sense, although planning is a complex component and, therefore, difficult to train, it is possible that the integrated form in which it has been trained in VIRTRAEL has achieved those small improvements that traditional stimulation has not achieved.

The findings of this study have a series of socio-sanitary implications since people with MCI who initiate cognitive stimulation by computer early reduce their risk of conversion to dementia [79] compared to those who do it late. It is also relevant that VIRTRAEL is a free tool available for early use by a large number of people with a suspicion of this type of deterioration. On the other hand, healthy seniors who carry out computerized cognitive stimulation programs and improve their reasoning abilities experience levels of independence for the instrumental activities of daily life, demonstrated in follow-ups up to 10 years [80]. VIRTRAEL is an easy program to be used by any older person, even from their own home, so it can contribute to the massive diffusion of cognitive stimulation “for all”. Finally, the innovative aspect of VIRTRAEL is its ecological approach. Computerized cognitive training offers the possibility of designing rich and diverse environments to trigger adaptive changes. VIRTRAEL exercises have been designed with this advantage in mind to favor the correspondence between the improvement of skills trained through the online platform and a better performance of daily activities. Our findings present an attractive alternative to traditional cognitive training, in which the premises of adjustment and individualization to the characteristics and specific needs of the person are paramount.

Among the limitations of the study is the small sample size, especially the group of standard cognitive stimulation, which limits the possibilities of generalization of the findings. Since this is a pilot study, the results indicate that it is possible to evaluate the efficacy of VIRTRAEL through a clinical trial with a larger sample size. In addition, the unbalanced size of the groups is a limitation. Although the equality of the variance of the groups minimizes the problem, for greater statistical power it is advisable to increase and balance the sample size in future studies. Another limitation derives from not having a follow-up of the improvements that allow knowing the stability of the cognitive gains reached by the older adults. In addition, no evaluations have been carried out on the transfer of improvements in the functioning of participants in activities of daily living or social participation. On the other hand, the merit of the study is to demonstrate that current technology makes it possible to translate the standard methodology of cognitive stimulation into a computer-based format and, in addition, to improve some of its key aspects. The effect of this achievement is an increase in the magnitude of cognitive improvements for older people. In order for older people to benefit from these advances, with VIRTRAEL they do not need to have previous computer skills, they only need to have access to a computer with an internet connection.

## 5. Conclusions

In conclusion, from a statistical and clinical point of view, the program of cognitive stimulation VIRTRAEL has a higher effectiveness than standard cognitive stimulation in improving the learning, verbal memory, working memory, and planning of older adults. This has great implications, even more so after the events of the COVID-19 pandemic, since VIRTRAEL is inexpensive and easily accessible to any older person, regardless of whether they live far from health care resources. The results should be interpreted with caution due to the limitations of this pilot study. Studies should be conducted to test the efficacy of the computerized cognitive platform in larger samples and in a population that meets the criteria for mild cognitive impairment, given that this disorder is considered a prodromal stage of Alzheimer’s disease.

## Figures and Tables

**Figure 1 brainsci-13-00684-f001:**
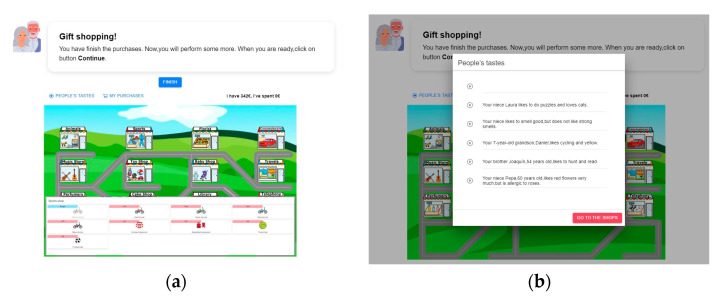
Example of exercise *Gifts Purchase*. It serves to improve planning skills (establishing goals, controlling implementation, and measuring results). The screen shows a shopping area, and the participant must buy a series of gifts for other people on account of each person’s listed preferences and within a limited budget. (**a**) is a screenshot of the main scenario, the stores that are in the city, along with an example of products that can be purchased in the sports store. (**b**) is an example of the people to buy a gift for and their tastes.

**Figure 2 brainsci-13-00684-f002:**
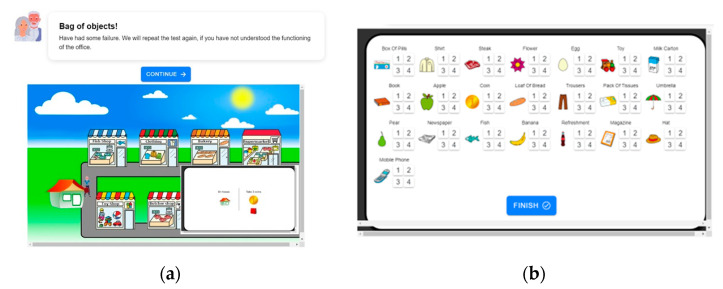
Example of exercise *Bag of Items*. It serves to improve the working memory based on a simulated walk through a neighborhood, in which the participant exchanges relevant objects in various local places. The user must memorize the objects that the person picks up and leaves along the route (in each of the establishments that he visits) in order to be able to indicate at the end of the exercise the objects that remain in the bag. (**a**) is a screenshot of the main scenario: the stores of a city and the house of the avatar guiding the exercise, together with an example of the objects that the avatar takes from home (coins) and carries in the bag. (**b**) is an example of a panel in which the participant must select the objects left in the bag after the tour of the city and the quantity of each of them.

**Figure 3 brainsci-13-00684-f003:**
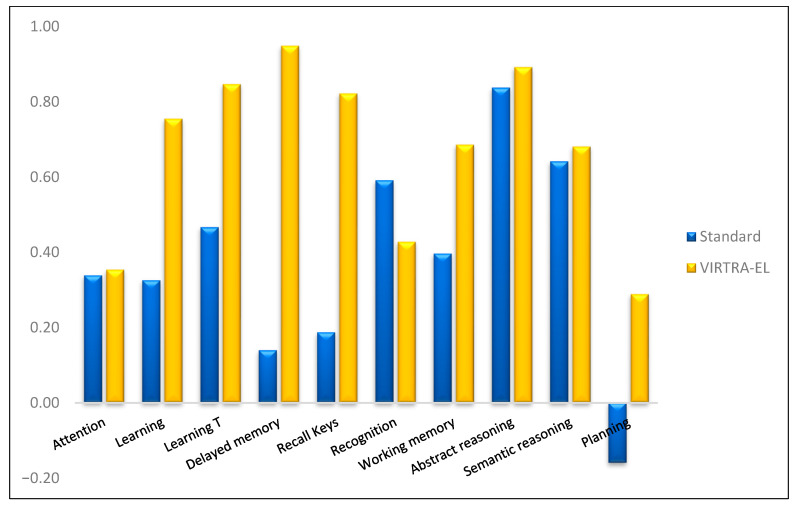
Mean clinical change in the VIRTRAEL and standard groups for the indices of each cognitive function.

**Table 1 brainsci-13-00684-t001:** Sociodemographics and general cognitive status of participants in both groups.

	VIRTRAEL Group (N = 55) $\bar{x}$ (SD)	Standard Group (N = 20) $\bar{x}$ (SD)	t (*p*)
Age	72.67 (6.46)	76.05 (7.58)	−1.911 (0.060)
Education (years)	6.62 (5.4)	4.95 (3.24)	1.626 (0.110)
Cognitive Reserve ^1^	7.87 (±4.39)	6.35 (±2.41)	1.902 (0.062)
MMSE ^2^	27.89 (1.65)	26.36 (1.68)	1.234 (0.221)

^1^ Cognitive Reserve = score on the cognitive reserve questionnaire; ^2^ MMSE = Mini-Mental State Examination [43,44].

**Table 2 brainsci-13-00684-t002:** Results of group differences in effectiveness and clinical change between VIRTRAEL and standard stimulation.

Cognitive Domain (Test)	Variable	VIRTRAEL Group		Standard Group		ANCOVA		Cohen’s *d* Pre-Post	
		Pre Mean (SD)	Post Mean (SD)	Pre Mean (SD)	Post Mean (SD)	*F*	*p*	VIRTRA-EL	Standard
**Attention** **(d2 ^1^)**	Concentration	107.49 (33.12)	119.20 (30.14)	82.50 (29.98)	92.65 (29.03)	2.738	0.102	0.354	0.339
Verbal Memory (HVLT-R ^2^)	Total learning	20.13 (4.44)	23.89 (3.90)	18.15 (3.96)	20 (4.27)	10.395	0.002	0.847	0.467
	Delayed recall	6 (2.13)	8.02 (2.10)	5.40 (2.50)	5.75 (2.40)	17.114	≤0.001	0.949	0.140
	Cued recall	7.04 (1.99)	8.67 (1.88)	6.70 (1.87)	7.05 (2.24)	12.187	0.001	0.822	0.188
	Recognition	10.44 (1.49)	11.07 (0.98)	10.20 (1.44)	11.05 (0.95)	0.004	0.948	0.428	0.592
Working Memory (WAIS ^3^)	L&N ^4^	6.55 (2.55)	8.20 (2.59)	5.20 (2.02)	6 (1.89)	8.272	0.005	0.686	0.397
Reasoning (WAIS)	Matrix	8.20 (4.22)	11.96 (4.75)	5.85 (2.21)	7.70 (2.06)	10.002	0.002	0.892	0.838
	Similarities	14.30 (4.52)	17.24 (5.08)	11.05 (2.72)	12.80 (3.14)	6.211	0.015	0.681	0.642
Planning (BADS ^5^)	Keys Search	7.18 (3.27)	7.96 (3.17)	6.30 (2.52)	5.90 (2.43)	5.612	0.021	0.239	−0.159 ^6^

^1^ d2Concentration: concentration index of the attention test d2; ^2^ HVLT-R: Hopkins Verbal Learning Test—Revised (total learning: sum of Trials 1–3; delayed recall: No. correct words in the delayed free recall trial; cued recall: recall facilitated by semantic keys; recognition: No. target words correctly recognized); ^3^ WAIS: Wechsler Adult Intelligence Scale III and IV; ^4^ L&N: Letters and Numbers; ^5^ BADS: Behavioral Assessment of the Dysexecutive Syndrome test battery. ^6^ The negative signs in Cohen’s d indicate that of the two moments being compared, the second performance is lower than the first.

## Data Availability

The authors have made the de-identified data on which the study conclusions are available at https://osf.io/2meku/ (accessed on 16 February 2023).

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
