# Peer review of "Effectiveness of Computerized Cognitive Training by VIRTRAEL on Memory and Executive Function in Older People: A Pilot Study"

_brainsci, 2023, doi:10.3390/brainsci13040684_

Round 1
Reviewer 1 Report
The authors are invited to complete the research hypotheses and expected outcomes of their research. In addition, it is essential to include information about the representativeness of the sample and validation of the tools used. The description of the results is very laconic, besides, the data in the second table could partly be presented in the form of a graph.
Author Response
Dear reviewer,
Thank you very much for your comments and recommendations to improve the article. We have taken into account what you have told us.
Comments and Suggestions for Authors:
The authors are invited to complete the research hypotheses and expected outcomes of their research. In addition, it is essential to include information about the representativeness of the sample and validation of the tools used. The description of the results is very laconic, besides, the data in the second table could partly be presented in the form of a graph.
Response: We have completed the hypothesis at the end of the introductory section by referring to the results we expect from the research. In addition, following your recommendation, we have included information on the representativeness of the sample by referring to a recent systematic review and meta-analysis that addresses this issue.
Information on validation of the tools used has been included in a paragraph directing the readers of this article to the documents where this information is contained.
The description of the results has also been expanded and we have added a graph in which part of the data from the second table is collected so that the information is more understandable.
We thank you again for your recommendations and we hope the manuscript meets the journal's high standards for publication after modifications are made.
Best regards.

Reviewer 2 Report
21 February 2023
Manuscript ID: brainsci-2254422
Type: Article
Title: ‘Effectiveness of a new free tool for computerized cognitive training in older people’ by Rute-Pérez S et al., submitted to Brain Sciences
Dear Authors,
Computerized cognitive training (CCT) allows individuals to access cognitive exercises from their own computers or mobile devices. The present research article, entitled ‘Effectiveness of a new free tool for computerized cognitive training in older people’ by Rute-Pérez and colleagues compared the effectiveness of VIRTRAEL (VIRtual TRAining for the ELderly) and traditional paper-and-pencil cognitive training in patients with cognitive impairment.
The main strength of this manuscript is that it addresses an interesting and timely question, reporting that VIRTRAEL is superior to the traditional training in improving verbal learning and memory, working memory, abstract and semantic reasoning, and planning abilities.
In general, I think the idea of this original article is interesting and the authors’ fascinating observations on this timely topic may be of interest to the readership of Brain Sciences. However, some comments as well as some crucial evidence should be included to support the author’s argumentation to improve its adequacy, its readability, and thus the quality of the manuscript.
Please consider the following comments:
1. Title: This is the most important section of the manuscript. Please present a concise and self-explanatory title stating the most important results and message of this study.
2. Abstract: I would like the authors to make as much effort for this section as for the rest of the manuscript. Please abridge the abstract to 200 words according to the guidelines of the journal (https://www.mdpi.com/journal/brainsci/instructions), proportionally presenting the background, the methods, the results, and the conclusion without subheadings.
The background should include the general background (one to two sentences), the specific background (two to three sentences), and current issue addressed to this study (one sentence), leading to the objectives. In this subsection I would like the authors to lay out basic information, problem statement, and the authors’ motivation to break off.
The methods should clarify the authors’ approach such as study design and variables to solve the problem and/or to make progress on the problem.
The results section must state the results in numbers and clarify their statistical significance. Are the results statistically significant? This subsection should close with a paragraph which puts the results into a more general context.
The conclusion should include one sentence describing the main result using such words like “Here we show”. The conclusion should write the potential and the advance this study has provided in the field and finally a broader perspective (two to three sentences) readily comprehensible to a scientist in any discipline.
3. Keywords: Please focus on choosing ten keywords from Medical Subject Headings (MeSH) (https://meshb.nlm.nih.gov/). I recommend adding ‘mania’ or equivalent as a keyword and use as many as possible in the title and in the first two sentences of the abstract
4. A graphical abstract that will visually summarize the main findings of the manuscript is highly recommended.
5. Introduction: The author needs to fully expand this section with up to 1000 words, introducing the main constructs of this study, which should be understood to a reader in any discipline and make persuasive enough to put forward the main purpose of current research the author has conducted and the specific purpose the authors has intended by this study. I would like to encourage the author to present the introduction starting with the general background in which main constructs can be neurodegenerative diseases affecting cognitive functions and other neurological disorders the authors would like to target on. Then, it proceeds to the specific background in which the main constructs can be traditional cognitive trainings, CCTs, and VIRTRAEL. Finally, the authors should present the current issue addressed to this study, leading to the objectives. Those main structures should be organized in a logical and cohesive manner. In this regard, a general overview of hallmarks of cognitive impairments, may include mitochondrial impairment as a common motif in neuropsychiatric presentation (https://doi.org/10.3390/cells11162607)’ and dissecting neurological and neuropsychiatric diseases: neurodegeneration and neuroprotection’ (https://doi.org/10.3390/ijms23136991). Also, I would recommend adding more information on neural substrates of cognitive dysfunction presented in mental illnesses as in ‘Functional interplay between central and autonomic nervous system’. This information may provide a better understanding of prefrontal cortex’s key role and how its disrupted function may contribute to irregular behavioral responses and therefore to the development of many cognitive dysfunctions that are common in neurological diseases (https://doi.org/10.3389/fnbeh.2022.998714).
6. Methods: I recommend opening this section with a short introductory paragraph regarding the study design and methodology. Also, I suggest citing more references to ensure the reliability and the integrity of evidence in the study design the authors have built and the methodology the authors applied to this study.
7. Results: I recommend that the authors close this section with a paragraph which put the results into a more general description.
8. Discussion: Generally, this section is well written; nevertheless, I would like the authors to complete this section with up to 1500 words, focusing on the following elements. Starting with the summary of the previous section (Results), the authors need to develop discussion on the potential of this study complementing as the extension of the previous work, the implication of the findings of this study, how this study could facilitate future research, the ultimate goal, the challenge, the knowledge and the technology necessary to achieve this goal, the statement about this field in general, and finally the importance of this line of research. It is particularly important to present its limit and its merit, and its potential translation of this study to clinical application.
9. Conclusion: I think that this section would benefit from a single paragraph presenting some thoughtful as well as in-depth considerations by the author as an expert to convey the take-home message, explaining the theoretical implication as well as the translational application of their research. I believe that it would be necessary to discuss theoretical and methodological avenues in need of refinement, as well as suggestions of a path forward in betterment of cognitive trainings.
10. References: Please follow the guidelines of the journal (https://www.mdpi.com/journal/brainsci/instructions) for reference style and add doi number.
Overall, the manuscript contains no figure, two tables, and 68 references. I believe that the manuscript may carry important value in studying the effectiveness of VIRTRAEL for patients with cognitive impairment. I hope the manuscript will meet the high standard of the journal for publication following the peer-review session. I am available for a new round of revision of this paper.
I declare no conflict of interest regarding this manuscript.
Best regards,
Reviewer
Author Response
Dear reviewer,
Thank you very much for your comments and recommendations to improve the article. We have taken into account what you have told us. Here are the changes we have made.
Comments and Suggestions for Authors:
- Title: This is the most important section of the manuscript. Please present a concise and self-explanatory title stating the most important results and message of this study.
Response: We have changed the title to make it more concise and state the results of the article. We believe that this way it will better convey the main idea of the paper.
- Abstract: I would like the authors to make as much effort for this section as for the rest of the manuscript. Please abridge the abstract to 200 words according to the guidelines of the journal (https://www.mdpi.com/journal/brainsci/instructions), proportionally presenting the background, the methods, the results, and the conclusion without subheadings.
The background should include the general background (one to two sentences), the specific background (two to three sentences), and current issue addressed to this study (one sentence), leading to the objectives. In this subsection I would like the authors to lay out basic information, problem statement, and the authors’ motivation to break off.
The methods should clarify the authors’ approach such as study design and variables to solve the problem and/or to make progress on the problem.
The results section must state the results in numbers and clarify their statistical significance. Are the results statistically significant? This subsection should close with a paragraph which puts the results into a more general context.
The conclusion should include one sentence describing the main result using such words like “Here we show”. The conclusion should write the potential and the advance this study has provided in the field and finally a broader perspective (two to three sentences) readily comprehensible to a scientist in any discipline.
Response: We have incorporated general and specific background information on the topic addressed in the study, which makes it easier to understand the objective we tried to achieve. In addition, the results have been presented including the statistical significance in order to make them clearer. In the conclusion subsection, we have included the words recommended by "Here we show" to highlight the potential of the article.
Following the guidelines of the journal the abstract has a total of 200 words.
- Keywords: Please focus on choosing ten keywords from Medical Subject Headings (MeSH) (https://meshb.nlm.nih.gov/). I recommend adding ‘mania’ or equivalent as a keyword and use as many as possible in the title and in the first two sentences of the abstract
Response: We have taken into account your suggestion on choosing keywords in Medical Subject Headings (MeSH). However, after a careful search of the terms, including the term "mania" that you recommended, we consider that the terms we propose are more appropriate for the subject matter of the article. We have not found any terms that really fit the main idea of the article.
- A graphical abstract that will visually summarize the main findings of the manuscript is highly recommended.
Response: A graphical abstract that visually summarizes the main findings of the manuscript has been sent to the editor for inclusion in the article.
- Introduction: The author needs to fully expand this section with up to 1000 words, introducing the main constructs of this study, which should be understood to a reader in any discipline and make persuasive enough to put forward the main purpose of current research the author has conducted and the specific purpose the authors has intended by this study. I would like to encourage the author to present the introduction starting with the general background in which main constructs can be neurodegenerative diseases affecting cognitive functions and other neurological disorders the authors would like to target on. Then, it proceeds to the specific background in which the main constructs can be traditional cognitive trainings, CCTs, and VIRTRAEL. Finally, the authors should present the current issue addressed to this study, leading to the objectives. Those main structures should be organized in a logical and cohesive manner. In this regard, a general overview of hallmarks of cognitive impairments, may include mitochondrial impairment as a common motif in neuropsychiatric presentation (https://doi.org/10.3390/cells11162607)’ and dissecting neurological and neuropsychiatric diseases: neurodegeneration and neuroprotection’ (https://doi.org/10.3390/ijms23136991). Also, I would recommend adding more information on neural substrates of cognitive dysfunction presented in mental illnesses as in ‘Functional interplay between central and autonomic nervous system’. This information may provide a better understanding of prefrontal cortex’s key role and how its disrupted function may contribute to irregular behavioral responses and therefore to the development of many cognitive dysfunctions that are common in neurological diseases (https://doi.org/10.3389/fnbeh.2022.998714).
Response: We have expanded the section from 556 to 868 words. We fully agree with you that the main constructs of this study should be included. Therefore, we have included relevant aspects regarding neurodegenerative diseases affecting cognitive functions and traditional cognitive training. We have also expanded the information in terms of the current topic addressed by this study leading to the objectives and the research hypothesis that we set out.
- Methods: I recommend opening this section with a short introductory paragraph regarding the study design and methodology. Also, I suggest citing more references to ensure the reliability and the integrity of evidence in the study design the authors have built and the methodology the authors applied to this study.
Response: We have opened the methods section with a brief introductory paragraph on the study design and methodology. In addition, we have added an explanatory sentence referring to the validation of the instruments applied to older people.
- Results: I recommend that the authors close this section with a paragraph which put the results into a more general description.
Response: Following your recommendation, we have included a paragraph to describe the results in a more general way. In addition, we have expanded the information and included a color graph to make the results clearer.
- Discussion: Generally, this section is well written; nevertheless, I would like the authors to complete this section with up to 1500 words, focusing on the following elements. Starting with the summary of the previous section (Results), the authors need to develop discussion on the potential of this study complementing as the extension of the previous work, the implication of the findings of this study, how this study could facilitate future research, the ultimate goal, the challenge, the knowledge and the technology necessary to achieve this goal, the statement about this field in general, and finally the importance of this line of research. It is particularly important to present its limit and its merit, and its potential translation of this study to clinical application.
Response: Thank you for your feedback on this section. It has been completed and we believe that all the elements you indicated in your commentary on this section are now present.
- Conclusion: I think that this section would benefit from a single paragraph presenting some thoughtful as well as in-depth considerations by the author as an expert to convey the take-home message, explaining the theoretical implication as well as the translational application of their research. I believe that it would be necessary to discuss theoretical and methodological avenues in need of refinement, as well as suggestions of a path forward in betterment of cognitive trainings.
Response: We have expanded this section with some considerations as to the final message we want to convey and the implications and scope it has. We have also suggested new avenues for expanding and improving knowledge about this computerized cognitive training program.
- References: Please follow the guidelines of the journal (https://www.mdpi.com/journal/brainsci/instructions) for reference style and add doi number.
Response: Following your recommendation, we have revised the references and adjusted some of them to the journal guidelines. In addition, we have added the doi number in all those that had it.
After following the recommendations for the different sections of the article, the number of references has increased. There are now 79 references.
We thank you again for your recommendations and we hope the manuscript meets the journal's high standards for publication after modifications are made.
Best regards.

Round 2
Reviewer 2 Report
1 March 2023
Manuscript ID: brainsci-2254422
Type: Article
Title: ‘Effectiveness of a new free tool for computerized cognitive training in older people’ by Rute-Pérez S et al., submitted to Brain Sciences
Dear Authors,
I am pleased to see that the authors took my comments seriously and have solved most of the issues I raised in the previous round of the peer-review session. Currently, the manuscript is a well written and nicely presented research article studying the effectiveness of VIRTRAEL (VIRtual TRAining for the ELderly) and traditional paper-and-pencil cognitive training in patients with cognitive impairment. I just leave here some comments that, I believe, help the authors improve the adequacy, the readability, and thus the quality of this manuscript to close my part of the review session.
Please consider the following comments:
1. Abstract: The authors present this section very nicely. I just suggest opening the conclusion with “Here we show” and adding the potential and the advance this study has provided in the field and finally a broader perspective (two to three sentences) readily comprehensible to a scientist in any discipline. Please mind the number of words. Personally, I would accept 200-220 words for an abstract.
2. Introduction: This section is nicely presented and has substantially improved. I just think this paper would benefit from mentioning the following topics such as dissecting neurological and neuropsychiatric diseases in terms of neurodegeneration and neuroprotection, and a better understanding of prefrontal cortex’s key role and how its disrupted function may contribute to irregular behavioral responses and therefore to the development of many cognitive dysfunctions that are common in neurological diseases (https://doi.org/10.3390/ijms23136991; https://doi.org/10.3389/fnbeh.2022.998714).
3. References: I would like the authors to follow the guidelines of the journal (https://www.mdpi.com/journal/brainsci/instructions) for reference style. I believe this careful but meticulous work would help assistant editors who work for us to publish high quality papers.
Overall, the manuscript contains one figure, two tables, and 68 references. I believe that the manuscript carries important value in studying the effectiveness of VIRTRAEL for patients with cognitive impairment. I am looking forward to seeing more papers written by the same authors.
Thank you.
I declare no conflict of interest regarding this manuscript.
Best regards,
Reviewer
Author Response
Dear reviewer,
Thank you very much for your comments, recommendations and the new opportunity to continue improving the article. We have taken into account your new comments and these are the modifications we have made.
Comments:
- Abstract: The authors present this section very nicely. I just suggest opening the conclusion with “Here we show” and adding the potential and the advance this study has provided in the field and finally a broader perspective (two to three sentences) readily comprehensible to a scientist in any discipline. Please mind the number of words. Personally, I would accept 200-220 words for an abstract.
Response: We have opened the conclusion with “Here we show” and added the potential and the advance that this study has provided in the field in two sentences. To adjust for word count, we have modified some sentences in the abstract.
- Introduction: This section is nicely presented and has substantially improved. I just think this paper would benefit from mentioning the following topics such as dissecting neurological and neuropsychiatric diseases in terms of neurodegeneration and neuroprotection, and a better understanding of prefrontal cortex’s key role and how its disrupted function may contribute to irregular behavioral responses and therefore to the development of many cognitive dysfunctions that are common in neurological diseases (https://doi.org/10.3390/ijms23136991; https://doi.org/10.3389/fnbeh.2022.998714).
Response: Following your recommendations we have mentioned the topic of neurodegeneration and neuroprotection and the role of the prefrontal cortex in the development of many cognitive dysfunctions. We thank you for the two references you have recommended. Finally, we have cited the first paper (https://doi.org/10.3390/ijms23136991) because we find it particularly interesting and related to the subject of the article.
- References: I would like the authors to follow the guidelines of the journal (https://www.mdpi.com/journal/brainsci/instructions) for reference style. I believe this careful but meticulous work would help assistant editors who work for us to publish high quality papers.
Response: We have reviewed and modified the references adding the Brain Sciences journal style to our Zotero manager. Consequently, all references have been modified to conform exactly to the specific style of the journal, two duplicate references have been deleted, and the new reference suggested by the reviewer in the introduction has been included (citation number 8). Finally, the manuscript contains 78 references.
We thank you again for your comments to improve the manuscript and hope that it now meets the journal's strict standards for publication.
Best regards.
